# Internal Ileal Diversion as Treatment for Progressive Familial Intrahepatic Cholestasis Type 1-Associated Graft Inflammation and Steatosis after Liver Transplantation

**DOI:** 10.3390/children9121964

**Published:** 2022-12-14

**Authors:** Anna M. Kavallar, Franka Messner, Stefan Scheidl, Rupert Oberhuber, Stefan Schneeberger, Denise Aldrian, Valeria Berchtold, Murat Sanal, Andreas Entenmann, Simon Straub, Anna Gasser, Andreas R. Janecke, Thomas Müller, Georg F. Vogel

**Affiliations:** 1Department of Paediatrics I, Medical University of Innsbruck, 6020 Innsbruck, Austria; 2Department of Visceral, Transplant and Thoracic Surgery, Medical University of Innsbruck, 6020 Innsbruck, Austria; 3Institute of Human Genetics, Medical University of Innsbruck, 6020 Innsbruck, Austria; 4Institute of Cell Biology, Biocenter, Medical University of Innsbruck, 6020 Innsbruck, Austria

**Keywords:** PFIC1, pediatric liver transplantation, steatosis, surgical biliary diversion

## Abstract

Background: Progressive Familial Intrahepatic cholestasis type I (PFIC1) is a rare congenital hepatopathy causing cholestasis with progressive liver disease. Surgical interruption of the enterohepatic circulation, e.g., surgical biliary diversion (SBD) can slow down development of liver cirrhosis. Eventually, end stage liver disease necessitates liver transplantation (LT). PFIC1 patients might develop diarrhea, graft steatosis and inflammation after LT. SBD after LT was shown to be effective in the alleviation of liver steatosis and graft injury. Case report: Three PFIC1 patients received LT at the ages of two, two and a half and five years. Shortly after LT diarrhea and graft steatosis was recognized, SBD to the terminal ileum was opted to prevent risk for ascending cholangitis. After SBD, inflammation and steatosis was found to be reduced to resolved, as seen by liver biochemistry and ultrasounds. Diarrhea was reported unchanged. Conclusion: We present three PFIC1 cases for whom SBD to the terminal ileum successfully helped to resolve graft inflammation and steatosis.

## 1. Introduction

Progressive Familial Intrahepatic cholestasis type I (PFIC1) is a rare congenital hepatopathy caused by bi-allelic mutations in the adenosine triphosphate (ATP)ase phospholipid transporting 8B1 (ATP8B1) gene, which encodes the aminophospholipid flippase FIC1 [1,2,3]. Presentation can be intermittent or chronical, and patients suffer from infantile cholestasis with low γ-glutamyltransferase (GGT) and pruritus. Cholestasis progresses to cirrhosis and end-stage liver disease [4]. Other clinical variants linked to ATPB1 expression are non-progressive forms such as benign recurrent intrahepatic cholestasis and intrahepatic cholestasis of pregnancy [5]. As FIC1 is expressed in various tissues other than liver, PFIC1 patients might also present with hearing loss, pancreatitis, hypoparathyroidism and diarrhea [6,7,8]. How loss of FIC1 function results in PFIC1 has not yet been entirely solved.

Thus far, treatment has been mostly symptomatic (e.g., nutritional, vitamin-supplementation and anti-pruritogenic agents) and often less effective. Surgical biliary diversion (SBD) procedures proved to have some effectiveness in alleviating pruritus and reducing the progression of liver fibrosis and disease [9,10,11,12,13,14]. SBD aims at diverting bile flow to either the body surface or to the colon, thereby bypassing the terminal ileum and reducing the amount of bile acids actively reabsorbed by the ileal apical sodium bile salt transporter/ ileal bile acid transporter (IBAT). For both internal or external SBD, various approaches have been demonstrated in recent years [15]. Recently, compounds inhibiting the IBAT have been developed and approved for the treatment of cholestatic pruritus [16,17].

However, some transplanted patients do not respond to available treatment and progress to end-stage liver disease, which ultimately requires liver transplantation (LT). Some patients suffer from aggravated diarrhea, increasing liver graft steatosis and inflammation with progressive fibrosis, which can even lead to graft loss [18,19,20,21,22,23]. Oral cholestyramine as a bile acid sequestrant drug is reported to be sufficient in treating persisting diarrhea after LT [24]. Biliary drainage or rerouting after LT via SBD can amend steatosis and graft injury [21,25,26,27]. Demonstrated techniques include external biliary drainage and diversion to the colon. 

Here, were report cases of three PFIC1 patients who developed post-LT graft steatosis that was successfully reverted with SDB to the terminal ileum.

## 2. Materials and Methods

### 2.1. Patient Cohort

All three patients underwent LT between May 2014 and October 2020 at the Department of Paediatrics I, Medical University of Innsbruck, Austria.

### 2.2. Genetic Analysis

Direct sequencing of all exons and flanking intronic sequences of ATP8B1 was carried out. ATP8B1 analysis using an Illumina TruSight One enrichment and variant calling by SeqNext software (current version of 04/2019, JSI medical systems, 77955 Ettenheim, Germany) identified the mutations. In all three cases, no other noteworthy mutations in PFIC-related genes were detected. 

Case 1: Direct sequencing of all exons and flanking intronic sequences of ATP8B1 (NCBI reference transcript NM_005603.6) revealed a paternally inherited heterozygous frameshift mutation, whereas no mutation was detected on the maternal allele at the time; however, this result was considered to confirm the clinical diagnosis of PFIC1, at the age of five and a half years. A maternally inherited, heterozygous deletion of exons 23–25 was later demonstrated by breakpoint-spanning PCR following the diagnosis of an affected brother (case 3, below). Case 2: Genetic testing was initiated at 9 months of age and consisted of Agilent SureSelect V6 (60 Mb) enrichment and Illumina-based exome sequencing as described (PMID: 34037727); however, an evaluation of PFIC-associated genes *ATP8B1*, *ABCB4*, *ABCB11*, *TJP2* and *MYO5B* did not reveal a pathogenic or likely pathogenic variant despite good coverage of these genes. A subsequent evaluation of the whole exome dataset did not identify any likely pathogenic, candidate variant either but indicated parental consanguinity and a disclosed a 7.8-Mb homozygous region around the *ATP8B1* locus. Given the clinical suspicion of PFIC1, the whole coding region of ATP8B1 was PCR-amplified in overlapping PCR fragments in cDNA synthesized from mRNA extracted from patient’s cultured skin fibroblasts and subjected to direct Sanger sequencing (primers and conditions are available from the authors upon request). Only after LT, genetic testing revealed a supposedly homozygous 179-bp insertion between exons 2 and 3, resulting in a frameshift and premature stop mutation in the ATP8B1 gene compatible with FIC1 deficiency (Table 1). Case 3: *ATP8B1* analysis using an Illumina TruSight One enrichment and variant calling by SeqNext software (current version of 04/2019, JSI medical systems, 77955 Ettenheim, Germany) identified the known, paternally inherited frameshift mutation and a maternally inherited deletion of exons 23–25 in a compound heterozygous state (Table 1). An analysis of additional genes associated with PFIC and monogenic hepatopathies was not conducted.

## 3. Case Reports

### 3.1. Case 1

A two-month-old male child, delivered vaginally at 40 weeks with a birthweight of 3900 g developed jaundice and pale stools from birth onward. Laboratory tests showed cholestasis with normal GGT. Etiological work up for neonatal jaundice was inconclusive and a liver biopsy was performed at the age of two months. Hepatocellular and canalicular cholestasis with mild portal-tract fibrosis was described. In the subsequent months, the patient developed progressive pruritus, poor growth as well as loss of hearing. PFIC diagnosis was suspected. At the age of five years, the patient underwent living donor liver transplantation (LDLT) using the left lateral segments from his mother, as a result of increasing fibrosis, deteriorating liver function and intractable pruritus. PFIC1 was confirmed genetically at the age of five and a half years (Table 1). Five months after LT, the patient developed rapid steatosis of the graft liver as well as progressive chronic diarrhea. At the age of nine years, an internal biliary diversion was performed to prevent further graft injury. Biliary reconstruction for LDLT in this case was performed via end-to-end cholodochostomy. For SBD, a hepaticojejunostomy to the terminal ileum was performed. Therefore, the terminal ileum was transected using a stapler device approximately 40 cm from the oral to the ileocecal valve. The hepaticojejunostomy was performed at the oral end of the aboral ileum loop followed by reconstruction with an end-to-side ileoileostomy close to the ileocecal valve (Figure 1A,B). Currently, after four-years follow-up, the patient still has issues with malabsorption of vitamin A, D, E and K and poor growth. He presents with normal liver tests (Table 1). He attends school and social activities without restrictions. Diarrhea was reported unchanged in frequency and consistency after SBD (Table 1). Ultrasound shows an inconspicuous graft with no sign of steatosis or fibrosis.

### 3.2. Case 2

A two-month-old female child was admitted to our hospital with jaundice and elevated conjugated bilirubin. She was born at term with birthweight of 3590 g. There was no history of liver disease within the family. The liver was mildly enlarged with an inhomogeneous parenchyma. There were no signs of portal hypertension. The gall bladder was of normal size and stools were yellow. Laboratory tests showed cholestasis with normal GGT. A liver biopsy was performed and revealed canalicular cholestasis without fibrosis. Causes of neonatal cholestasis were excluded earlier but PFIC1 did not succeeded as proof at first sight. Within the following months, the girl developed increasing pruritus and failure of thrive. Her clinical condition worsened with decreased fibrinogen and signs of increasing hepatic fibrosis and portal hypertension. At the age of three years, the patient underwent LT with a deceased donor full-size organ. Biliary reconstruction was again performed with an end-to-end cholodochostomy. Only after LT, the genetic testing revealed a complex homozygous insertion resulting in a frameshift mutation in the ATP8B1 gene compatible with FIC1 deficiency (Table 1). After initial improvement of hepatic transaminases post-LT, an increase in alanine-aminotransferase (ALT) and aspartate-aminotransferase was noted and steatosis was shown by ultrasound 6 months after LT. Two and a half years after the LT procedure, a liver biopsy showed severe steatosis with mild fibrosis and focal mild inflammation. To avoid progressive fibrosis and subsequent graft loss, SBD to the terminal ileum was performed. Due to the duct-to-duct biliary reconstruction during LT, similar to the first case, a hepaticojejunostomy to the terminal ileum was performed and continuity was achieved with an end-to-side ileoileostomy close to the ileocecal valve (Figure 1C,D). In this case, ileal transection was performed approximately 15 cm from the oral to the ileocecal valve. Currently, one year after SBD, the patient continues to have normal liver function tests and reports improved diarrhea. Ultrasound demonstrates homogenous parenchyma without signs of steatosis.

### 3.3. Case 3

The younger brother of patient 1 was born at term after an uncomplicated pregnancy. He was admitted to our hospital because of scleral jaundice at the age of two months. Laboratory tests showed cholestasis with low GGT. An ultrasound examination demonstrated a regular sized liver and spleen. Given his brother’s PFIC1 diagnosis, liver biopsy and genetic analysis were performed immediately. Sequencing of ATP8B1 identified compound heterozygous mutations (Table 1). Liver histopathology revealed canalicular cholestasis without fibrosis. Similar to his brother, the patient developed pruritus that was only partially responsive to anti-pruritogenic therapy. At the age of seven months, an ultrasound showed enlargement of the liver. With the age of six months, he developed progressive diarrhea. At the age of 21 months, LDLT of the left lateral segments was performed on his father to address acute chronic liver failure and intractable pruritus. Biliary reconstruction during LDLT was performed with a Roux-en-Y loop. Steatosis was shown 4 months after LT and SBD was performed at the age of three years (13 months after LT). In this case, the existing biliary Roux-en-Y loop was transected at the level of the choledochojejunostomy using a stapling device. The cut end was then anastomosed in an end-to-side fashion to the terminal ileum in close proximity to the ileocecal valve (Figure 1E,F). Total length from hepaticojejunostomy to the ileocecal valve was approximately 18 cm. The initial normalization of hepatic transaminases changed to an undulant course over in subsequent months. The patient developed cholangitis with stenosis of the bile duct’s ostium, and a percutaneous transhepatic cholangio-drainage was placed and antibiotic treatment was initiated. Microbiological analysis revealed an infection with Escherichia coli. He caught up in height and weight from initial dystrophia to be in the 25th percentile. Six months after SBD, liver tests and an ultrasound did not show signs of steatosis or fibrosis. There was no significant improvement in diarrhea.

## 4. Discussion

PFIC1-related end-stage liver disease necessitates LT. However, the multisystemic nature and complex pathophysiology of PFIC1 yield new risks and symptoms for PFIC1 patients after LT. Particularly, graft steatosis, inflammation and progressive fibrosis may lead to graft loss and the need for re-transplantation. To prevent graft injury, SBD after LT has proven to be effective [21,25,26,27]. We present three patients, for whom SBD to the terminal ileum proved to be effective. External diversion was declined by parental decision in all three patients. The risk of ascending cholangitis after diversion of the biliodigestive loop to the colon, where the number of commensal gut microbiota is very high, was considered unreasonable in the post-LT setting. Therefore, SBD to the terminal ileum was opted for. This led to a decline in liver injury seen by a decrease of hepatic transaminases and an improvement in steatosis, as seen in the ultrasounds (Table 1). One limitation of the presented cases is the missing follow-up liver biopsy after post-LT SBD. However, the procedure did not strongly impact the reported frequency of diarrhea, where improvement was seen in some cases of diversion to the colon [25,26,27]. Still, to address this properly, a systematic study of more patients would be desirable. To rule out diarrhea related to pancreatic insufficiency as a possible clinical manifestation of PFIC1, it is recommended to perform fecal elastase testing. The patients presented here have normal to slightly reduced levels but pancreatic insufficiency has to be further considered.

Conceivably, the reconstituted and likely increased amount of bile acids in the intestinal lumen of PFIC1 patients after LT drive liver steatosis, inflammation and diarrhea should be considered. One hypothesis would be that the increased amount of bile acids secreted after LT into the intestine drives chologenic diarrhea and subsequently alters intestinal microbial composition. Conceivably, increased translocation of bacterial molecules to the liver via the portal vein might occur, resulting in graft steatosis and inflammation. However, the exact underlying pathophysiology remains to be elucidated [28,29]. As FIC1 is highly expressed [13], this might corroborate this hypothesis but the entire impact of FIC1 loss on enterocyte and mucosal homeostasis is not fully understood. SBD reduces the amount of bile acids in certain parts of the intestine and thereby alleviates graft inflammation and steatosis.

Furthermore, total parenteral nutrition is associated with hepatic biochemical and morphologic changes and can cause steatosis. In our cases, supplementary parenteral nutrition was provided less than seven days immediately after transplantation and we can exclude parenteral nutrition as an additional cause of steatosis. Additionally, the immunosuppressive regimen used can cause diarrhea or steatosis. In all three cases, tacrolimus was used as the main immunosuppressive therapy. However, patient 1 received mycophenloate mofetil (MFF) for two months and patient 3 for 10 months. Withdrawal of MFF did not influence diarrhea. As subacute rejection was suspected in patient 1 and 2, two courses of prednisolone pulse therapy was administered in case 1 and one course in case 3. Nonetheless, steatosis had been identified before prednisolone pulses in both patients.

Apparently, diversion to the distal part of the terminal ileum, as demonstrated in our cases, sufficiently reduces reabsorbed bile acids and thereby alleviates hepatic inflammation and steatosis. Yet, fat-soluble vitamin deficiencies might increase, as was seen in all three cases (Table 1). On the other hand, our approach seems inefficient in controlling post-LT diarrhea, as it was found to be improved only in patient 2. Different techniques of SBD have been reported [21,25,26,28,30].

Due to the limited number of affected patients, the published literature is limited to case reports or case series and correct timing and type of diversion is still unknown. SBD can be performed both as external or internal biliary diversion and procedures can be divided into partial and total biliary diversions that can be performed either pre- or post-LT [26]. Given the clinical courses and presentation of our patients, SBD was performed after LT in all cases. Partial diversion is done by rerouting bile from an existing gale bladder or bile duct either to the body surface (partial external biliary diversion; PEBD) or to the colon (partial internal biliary diversion; PIBD) while preserving bile flow in the common bile duct [30]. This can be achieved with drain placements or via enteric conduits [21,31]. Total diversion involves complete rerouting of produced bile either externally or internally (TIBD). In order to better facilitate partial biliary diversion post-LT, the gall bladder may be preserved in case of full-size grafts, though reports on this particular procedure are missing and are limited to the pre-LT setting. One case of PIBD to the colon using a side-to-side cholecdochojejunostomy in a patient after cadaveric split liver transplant was reported by Alrabadi et al. [21]; however, the patient developed ascending cholangitis and liver abscess. Alternatively, total biliary diversion is an option for progressive graft inflammation and refractory diarrhea post-LT. Despite reports of satisfactory outcome of external biliary diversion during or post-LT [21,31] added burden of the necessary stoma was met by parental rejection in all of our cases. TIBD is another possible technique that has been described in two cases after LT allowing for biliary diversion without the need for an external stoma [25,26]. In all instances, a small bowel (jejunum) conduit was used to divert bile salts to the colon. Authors addressed their reasoning for usage of a rather long segment and a small jejunocolostomy with non-resorbable sutures to decrease the risk of ascending infections. In both instances, no ascending infectious complications were reported and graft function remained stable in the case performed concomitantly to LT and markedly improved in the other [25,26].

To minimize the risk of ascending infection, we hypothesized that an intact ileocecal valve serving as barrier to the heavily colonized colon would significantly reduce the risk of infection in SBD patients after LT. We then explored the use of the terminal ileum in patients without a preexisting hepaticojejunostomy. Though conceptually controverse, we favored this approach in our first two cases to avoid the need for multiple small bowel anastomoses (as would have been necessary with a jejunal segment interponant). After our first case, where we used a rather long ileal segment, we revised our surgical technique and reduced the length of the biliary loop in the subsequent two cases. This resulted in a similar reduction of hepatic inflammation and steatosis. In the last case, the original choledochojejunostomy was transected and rerouted directly in the terminal ileum before the ileocecal valve. This was the only patient who developed ascending cholangitis with stenosis of the hepaticojejunostomy and need for interventional treatment. To prevent ascending cholangitis, a longer Roux-en-Y loop could be advisable.

In our experience with three cases, the rerouting of bile in the terminal ileum improved hepatic inflammation as evident by laboratory tests and imaging and at a follow-up between 6 months and 4 years. Diarrhea improved in one of our patients and ascending cholangitis occurred in another one. Finally, it is unclear whether the length of the biliary loop contributed to this observation.

The growing body of literature highlighting the post-LT course of PFIC1 patients suggests that SBD should be performed during LT to prevent graft steatosis and the necessity of a second surgical procedure, as post-LT complications in PFIC1 seem likely. Both external and internal biliary division are possible strategies for SBD. A plausible alternative might be the application of an IBAT inhibitor [32]. The impact on the enterohepatic circulation and reduction of bile acids in the small intestinal lumen might be comparable. Future studies will explore the potential of IBAT inhibition.

## Figures and Tables

**Figure 1 children-09-01964-f001:**
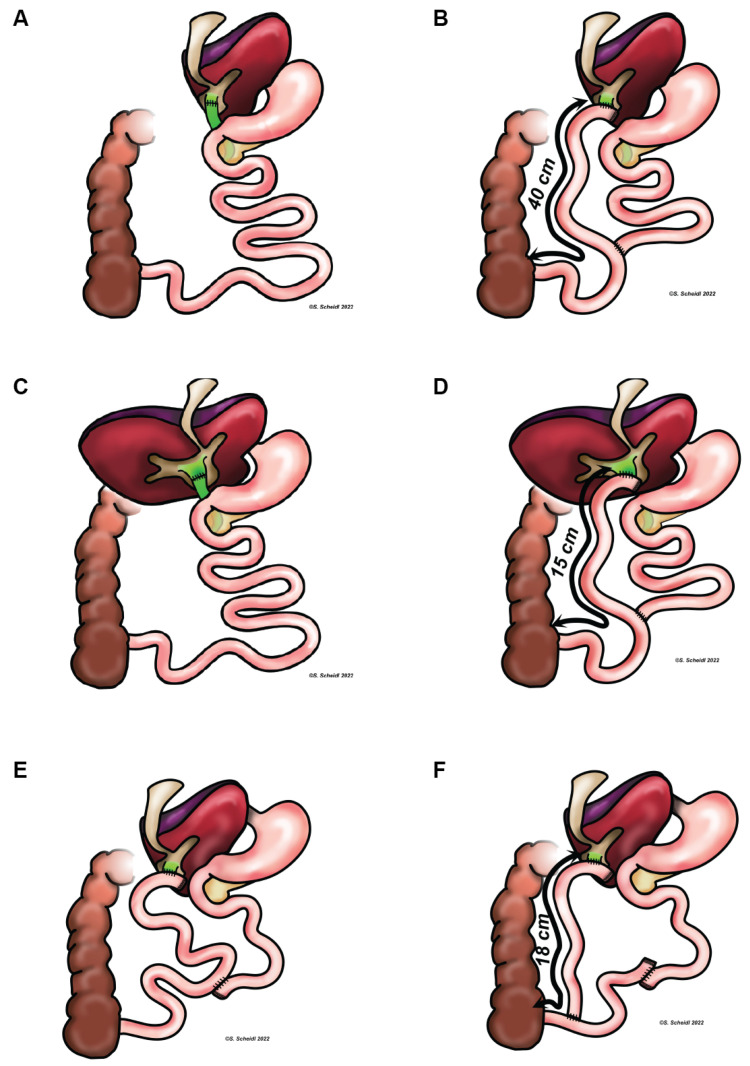
Biliary reconstruction after liver transplantation (**A**,**C**,**E**) and corresponding anatomy after internal biliary diversion (**B**,**D**,**F**). Anatomical situation after liver transplantation and before (**A**) as well as after (**B**) biliary diversion in case 1, case 2 (**C**,**D**), and case 3 (**E**,**F**), respectively.

**Table 1 children-09-01964-t001:** Clinical and genetic findings in PFIC1 patients.

		Patient 1	Patient 2	Patient 3
general	sex	male	female	male
	age first symptoms	2 mo	2 mo	1 mo
	age at diagnosis	5.5 yrs	3 yrs	2 mo
	*ATP8B1* variant 1	c.1214_1215del *	c.181_182ins179 *	c.1214_1215del *
	*ATP8B1* variant 2	c.(2931 + 1_2932-1)_ (3400 + 1_3401-1)del *	c.181_182ins179 *	c.(2931 + 1_2932-1)_ (3400 + 1_3401-1)del *
pre-LT	indication	acute liver failure, INR 3.6, fibrinogen 195	acute liver failure, INR 1.5, fibrinogen 87	INR 2.5, fibrinogen 269, pruritus
	age	5 yrs	2.5 yrs	21 mo
	graft	LDLT (mother)	DD fullsize	LDLT (father)
	biliary anastomosis	E/E	E/E	H/J
	vitamin deficiency	D, E, K	/	D, K
	glycemia (mg/dL)	76	81	96
	cholesterol (mg/dL)	93	151	133
	triglycerides (mg/dL)	126	233	196
post-LT	AST (U/L)	488	208	326
	ALT (U/L)	242	128	262
	GGT (U/L)	60	27	72
	serum bile acids (μmol/L)	10.3	/	/
	glycemia (mg/dL)	84	78	98
	cholesterol (mg/dL)	72	76	101
	triglycerides (mg/dL)	45	68	71
	US	hepatic steatosis, portal vein stenosis 4 mm, splenomegaly	hepatic steatosis, splenomegaly	hepatic steatosis, splenomegaly
	ultrasound elastography	-	4.93 kPa	10.5
	Bx	steatosis (85%) with periportal spearing, no steatohepatitis	Steatosis (>90%) with focal and mild steatohepatitis	mild to moderate portal-tract fibrosis, mild lobular chronic inflammation, steatosis (20%)
	diarrhea (per day)	5–8	4–7	10–12
	vitamin deficiency	D, A, K	D, K	D, K
post-LT SBD	age	9 yrs	6 yrs	3 yrs
	AST (U/L)	47	47	37
	ALT (U/L)	74	32	60
	GGT (U/L)	19	46	78
	serum bile acids (μmol/L)	11.8	8.9	16.6
	glycemia (mg/dL)	79	69	78
	cholesterol (mg/dL)	113	106	101
	triglycerides (mg/dL)	89	107	83
	US	hyperechogenic parenchyma, no splenomegaly	homogeneous parenchyma, no splenomegaly	homogeneous parenchyma, no splenomegaly
	ultrasound elastography	2.9 kPa	3 kPa	1.75 kPa
	diarrhea (per day)	6–8	3	8–10
	vitamin deficiency	A, D, E, K	A, D, E, K	D, K
	fecal elastase (μg/g)	293	436	162

Patient characteristics and synopsis of liver function tests, ultrasound (US) findings and biopsy (Bx) results prior to liver transplantation (LT), post-LT and post-LT SBD. ALT: alanine-aminotransferase, AST: aspartate-aminotransferase, H/J: hepaticojejunostomy, DD: deceased donor, E/E: end to end, GGT—γ-glutamyltransferase, LDLT: living donor liver transplantation. * Three distinct and novel *ATP8B1* variants were detected in these patients; these *ATP8B1* variants all cause frameshifts and premature stop codons predicting nonsense-mediated mRNA decay, and are all classified as pathogenic by ACMG criteria.

## Data Availability

The data presented in this study are available on request from the corresponding author.

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
