# Peer review of "Internal Ileal Diversion as Treatment for Progressive Familial Intrahepatic Cholestasis Type 1-Associated Graft Inflammation and Steatosis after Liver Transplantation"

_children, 2022, doi:10.3390/children9121964_

Round 1

Reviewer 1 Report

Nice work - I would only suggest minor changes:

1. Introduction: please describe the concept of SBD a bit more detailed in the introduction, so that the reader understands the surgical concept without reading the full article

2. I would use "hepaticojejunostomy" instead of "biliodigestive anastomosis"

Author Response

The authors in this exciting paper describe three cases of children undergoing internal ileal Nice work - I would only suggest minor changes:

  1. Introduction: please describe the concept of SBD a bit more detailed in the introduction, so that the reader understands the surgical concept without reading the full article

Reply: We agree with the reviewer and have elaborated on the concept of SBD in the introduction.

  1. I would use "hepaticojejunostomy" instead of "biliodigestive anastomosis"

Reply: We have now have put "hepaticojejunostomy" instead of "biliodigestive anastomosis", as suggested.

Reviewer 2 Report

In this case series, the authors present in-depth three PFIC1 cases affected by the recurrence of diarrhoea and steatosis after transplantation for whom SBD to the terminal ileum successfully helped to resolve graft inflammation and fatty liver.

Only a few minor revisions are needed before considering the manuscript for publication in Children.

One observation for all three patients: drug therapy must be included in all three cases.

- How many took UDCA after the transplant, and at what dose?

- What was the immunosuppressive therapy used? Were there patients taking mycophenolate mofetil, and at what dose? Did the children take cortisone?

This information is essential since MMF can cause diarrhoea and steroid steatosis.

In the discussion, this reference could be cited and be commented on:

- Panayotis Lykavieris, Saskia van Mil, Danièle Cresteil, Monique Fabre, Michelle Hadchouel, Leo Klomp, Olivier Bernard, Emmanuel Jacquemin,

Progressive familial intrahepatic cholestasis type 1 and extrahepatic features: no catch-up of stature growth, exacerbation of diarrhea, and appearance of liver steatosis after liver transplantation, Journal of Hepatology, Volume 39, Issue 3, 2003, Pages 447-452, ISSN 0168-8278, https://doi.org/10.1016/S0168-8278(03)00286-1.

Author Response

In this case series, the authors present in-depth three PFIC1 cases affected by the recurrence of diarrhoea and steatosis after transplantation for whom SBD to the terminal ileum successfully helped to resolve graft inflammation and fatty liver.

Only a few minor revisions are needed before considering the manuscript for publication in Children.

One observation for all three patients: drug therapy must be included in all three cases.

- How many took UDCA after the transplant, and at what dose?

Reply: None of our patients received UDCA after transplantation.

- What was the immunosuppressive therapy used? Were there patients taking mycophenolate mofetil, and at what dose? Did the children take cortisone?

This information is essential since MMF can cause diarrhoea and steroid steatosis.

Reply: We agree that this is important information! In all three cases tacrolimus was used as the main immunsuppressive therapy. Patient 1 received MMF for 2 months and patient 3 for 10 months. However, diarrhea did not change after withdrawal of MFF.

Due to rejection two courses of prednisolone pulse therapy was administered in case 1 and one in case 3. However, steatosis has been identified in both patients before the administration of prednisolone.

In the discussion, this reference could be cited and be commented on:

- Panayotis Lykavieris, Saskia van Mil, Danièle Cresteil, Monique Fabre, Michelle Hadchouel, Leo Klomp, Olivier Bernard, Emmanuel Jacquemin,

Progressive familial intrahepatic cholestasis type 1 and extrahepatic features: no catch-up of stature growth, exacerbation of diarrhea, and appearance of liver steatosis after liver transplantation, Journal of Hepatology, Volume 39, Issue 3, 2003, Pages 447-452, ISSN 0168-8278, https://doi.org/10.1016/S0168-8278(03)00286-1.

Reply: We have already cited this reference in the discussion (reference 30), as it is important.